# Fiber Spinning of Polyacrylonitrile Terpolymers Containing Acrylic Acid and Alkyl Acrylates

Ivan Yu. Skvortsov [1,*], Mikhail S. Kuzin [1], Andrey F. Vashchenko [1], Roman V. Toms [1,2], Lydia A. Varfolomeeva [1], Elena V. Chernikova [1,3], Gulbarshin K. Shambilova [4,5] and Valery G. Kulichikhin [1]

1   A. V. Topchiev Institute of Petrochemical Synthesis, Russian Academy of Sciences, 119991 Moscow, Russia; gevahka60@mail.ru (M.S.K.); vaanfo@yandex.ru (A.F.V.); toms.roman@gmail.com (R.V.T.); varfolomeeva.lidia@mail.ru (L.A.V.); chernikova_elena@mail.ru (E.V.C.); klch@ips.ac.ru (V.G.K.)
2   M. V. Lomonosov Institute of Fine Chemical Technologies, MIREA—Russian Technological University, 119571 Moscow, Russia
3   Department of Chemistry, Moscow State University, 119991 Moscow, Russia
4   S. Utebayev Atyrau Oil and Gas, University, md. Privokzalnyi, st. Baimukhanov, 45A, Atyrau 060027, Kazakhstan; shambilova_gulba@mail.ru
5   Department of Chemistry and Chemical Technology, Kh. Dosmukhamedov Atyrau University, Studenchesky Ave., 1, Atyrau 060011, Kazakhstan
*   Correspondence: amber5@yandex.ru

**Abstract:** Terpolymers of acrylonitrile with acrylic acid and alkyl acrylates, including methyl-, butyl-, 2-ethylhexyl-, and lauryl acrylates, were synthesized using the reversible addition–fragmentation chain transfer method. In this study, the focus was on the investigation of the impact of different monomer addition methods (continuous and batch) on both the rheological behavior of the spinning solutions and the mechanical properties of the resulting fibers. Our findings revealed that the method of monomer addition, leading either to non-uniform copolymers or to a uniform distribution, significantly influences the rheological properties of the concentrated solutions, surpassing the influence of the alkyl-acrylate nature alone. To determine the optimal spinning regime, we examined the morphology and mechanical properties at different stages of fiber spinning, considering spin-bond and orientation drawings. The fiber properties were found to be influenced by both the nature and introducing method of the alkyl-acrylate comonomer. Remarkably, the copolymer with methyl acrylate demonstrates the maximum drawing ratios and fiber tensile strength, reaching 1 GPa. Moreover, we discovered that continuous monomer addition allows for reaching the higher drawing ratios and superior fiber strength compared to the batch method.

**Keywords:** co-polyacrylonitrile; RAFT process; comonomer sequence; alkyl acrylate; mechanotropic spinning; fibers; mechanical properties

## 1. Introduction

Polyacrylonitrile (PAN) is widely used as a polymer for producing high-strength carbon fibers [1–3]. Although the process of PAN fiber carbonization has been studied for a long time [4–7], the optimal copolymer composition required for producing high-quality precursor fibers and high-strength carbon fibers from them remains uncertain. This uncertainty is derived from the need to solve a set of complex problems, starting from copolymer synthesis and polymer processing into precursor fibers to finishing of thermal oxidative stabilization and carbonization/graphitization [8,9]. The carbon fiber properties depend on the conditions, regimes, and parameters chosen to be applied at each stage [2,10].

Acrylonitrile homopolymer is typically unsuitable for producing textile or precursor fibers due to the solution gelation because of the strong interaction of nitrile groups, making it impossible to achieve a high degree of orientation that is critical for producing high-quality fibers [11,12]. Additionally, cyclization in homopolymers on the thermal oxidation

stage occurs within a narrow temperature range with a high exo-effect, making it impossible to reach a uniform distribution of poly-conjugated bonds [13–15].

To solve these challenges, precursor fibers are typically prepared from double or ternary copolymers, depending on the desired properties of the final fiber. To accelerate the formation of PAN poly-conjugated bonds, an acid comonomer is added, typically itaconic acid [16,17] or acrylic acid (AA) [7,18]. Alkyl acrylate comonomers (such as methyl acrylate (MA)) are often used as plasticizers [19,20]. Comparative studies on the effect of comonomer additives on the rheological behavior and thermal properties of the polymer indicate that the chosen strategy simplifies the processing of the polymer into fibers and improves their thermal and mechanical properties [21–23], confirming its effectiveness.

Reversible addition–fragmentation chain transfer (RAFT) polymerization is a relatively new method for polymer synthesis [24,25]. It enables working with a broad spectrum of comonomers, controlling the molecular weight distribution and distribution of comonomers along the chain, which can significantly influence the properties of the resulting polymers. These parameters are essential for the fiber spinning and the thermal behavior of the polymer during thermal cyclization, which are crucial for obtaining high-quality textile and precursor fibers [26–28].

The influence of the synthesis method and the sequence of adding AA to the reaction syrup of binary copolymers of acrylonitrile (AN) with AA has been studied [29–31]. In the case of radical polymerization, the rheological properties of solutions show little change depending on the method of loading the components. However, in the case of RAFT synthesis, the viscoelastic characteristics of both dilute and concentrated solutions are dependent on features of the synthesis procedure. Additionally, solutions of polymers synthesized by RAFT synthesis, with similar molecular weight and concentration, have lower viscosity compared to polymer solutions synthesized by classical radical polymerization.

In [32], the role of the MA fragment in the polymer was analyzed by studying the rheological properties of high-molecular-weight binary and ternary copolymers with similar molecular weights. It was found that in the concentration range above 5%, terpolymers' solutions exhibited a relative decrease in the elastic modulus, and a decrease in the pseudo-plasticity index due to changes in concentration and temperature was more significant in the terpolymer compared to the copolymer, which suggests that MA acts as a component that reduces intermolecular interactions.

The influence of the synthesis method on solution properties of ternary copolymers of AN, AA, and alkyl acrylates (AAc), obtained by either RAFT or radical polymerization, was studied [7]. There was no solution rheology difference at high concentrations but significant differences in the behavior of dilute solutions was observed due to changes in the interaction of the polymer with the solvent. Continuing this work, a series of ternary copolymers based on AN, AA, MA was synthesized by the RAFT method, varying the sequence of component loading. Thermal behavior analysis showed that the cyclization occurs over a wider temperature range with the increasing length of the alkyl substituent in alkyl acrylate, with the maximum observed for copolymers containing lauryl acrylate fragments [33]. The significance of the sequence of component addition was studied in a pair of terpolymers, also based on AN, AAc, and methyl acrylate synthesized by RAFT, revealing that a uniform distribution of comonomers along the chain promotes better polymer dissolution, yielding less viscoelastic solutions at concentrated region. The nonuniform copolymer demonstrated the best ability for orientational and plasticizing stretching that led to spinning much stronger fibers [34].

The goal of this study was the investigation of the influence of the addition of a broad range of ternary copolymers of PAN with various alkyl acrylates, including methyl-, butyl-, ethylhexyl-, and lauryl acrylate, on the solutions rheology in a novel mechanotropic fiber-spinning process that does not use coagulation baths and PAN fiber properties. The synthesis was carried out in two ways, i.e., by producing statistically homogeneous copolymers and copolymers of predominantly nonuniform structure, as described in [33], and their synthetic conditions and properties were reported. In the first stage, we analyzed the

effect of the alkyl substituent on the rheological properties of dilute solutions, which is essential for understanding the interaction of the polymer with a solvent and concentrated systems used for fibers spinning. In the second stage, we obtained a series of fibers by mechanotropic spinning and studied their mechanical properties. The results obtained bring us closer to understanding the role of the alkyl substituent in nature and compositional homogeneity of PAN in the production of high-strength fibers, including precursors for carbon fiber.

## 2. Materials and Methods

### 2.1. Materials

Ternary copolymers were synthesized using the RAFT method [35]. The series of copolymers were obtained with a molar monomer ratio of AN:AA:AAc of 93:7:3.5 and 98:2:10. All monomers were produced by Acros Organics (Fisher Scientific GmbH, Schwerte, Germany) except for LA (Sigma Aldrich, Merck Group, St. Louis, MO, USA). In one case, AA and AAc were continuously introduced into the reaction mixture containing AN, initiator (Sigma Aldrich, Merck Group, St. Louis, MO, USA), and RAFT agent (Sigma Aldrich, Merck Group, St. Louis, MO, USA), resulting in a polymer with a nonuniform structure (Samples MAc, BAc, LAc). In the second case, all monomers were added at the same time, resulting in a copolymer with a uniform distribution of monomer units along the chain (samples MAb1, MAb2, BAb1, BAb2, LAb1, and EHAb1). The synthesis conditions are described in detail in [33], and the polymer characteristics are presented in Table 1.

**Table 1.** Copolymer characteristics, where c = continuous comonomer addition method, b = batch comonomer addition method; 1 = high MW, 2 = low MW.

| Sample | Đ | $M_n \times 10^{-3}$ | Alkyl Acrylate | Comonomer Addition Method | $f_{AH}$, mol.% | $f_{AH}$, mol.% |
|---|---|---|---|---|---|---|
| MAc | 1.5 | 50 | | Continuous | 88.4 | 72.2 |
| MAb1 | 1.4 | 76 | Methyl- | Batch | 89.8 | 91.3 |
| MAb2 | 1.7 | 53 | | Batch | 88.4 | 89.5 |
| BAc | 1.6 | 85 | | Continuous | 88.4 | 72.4 |
| BAb1 | 1.4 | 95 | Butyl- | Batch | 89.8 | 92.3 |
| BAb2 | 1.7 | 61 | | Batch | 88.4 | 85.5 |
| LAc | 1.7 | 45 | Lauryl- | Continuous | 88.4 | 84.2 |
| LAb1 | 1.7 | 101 | | Batch | 89.8 | 91.2 |
| EHAb1 | 1.4 | 75 | Ethylhexyl- | Batch | 89.8 | 92.6 |

### 2.2. Methods

#### 2.2.1. Preparation of Solutions

Polyacrylonitrile powder and dimethyl sulfoxide (DMSO) of 99.5% from Ekos-1 (Moscow, Russia) were mixed in glass vials with hermetically sealed lids. Different dissolution methods were employed based on the polymer concentration:

- For solutions with a polymer concentration of 1–5 wt%, the mixture was stirred for 24 h at 50 °C using a magnetic stirrer. For concentrations below 1 wt%, the solution was sequentially diluted inside a Ubbelohde capillary viscometer at 25 °C to determine the intrinsic viscosity.
- Highly viscous solutions with a polymer content greater than 5 wt% were prepared using a paddle mixer with a J-shaped rotor. Mixing was conducted at 60 rpm for 24 h at 70 °C.
- High-viscosity spinning solutions with concentrations above 20 wt% were prepared using a rotor speed of 10 rpm for 72 h at 70 °C.

### 2.2.2. Rheology

The rheological behavior of the solutions was studied using shear and oscillatory deformation modes on a HAAKE MARS 60 rheometer (Thermo Fisher Scientific, Karlsruhe, Germany) at temperatures ranging from 20 to 70 °C, with a cone-plate system having an angle of 1° between the cone and the plate. A 60 mm diameter rotor was used for solutions with concentrations less than 10%, while a 25 mm diameter rotor was used for concentrations greater than 10%. The flow curves were obtained in the shear rate range of $10^{-1}$–$10^3$ s$^{-1}$, and the frequency dependences of the storage and loss moduli were obtained in the linear viscoelastic region at angular frequencies ranging from 0.6 to 628 rad/s. The intrinsic viscosities of the solutions were determined at 25 °C using an Ubbelohde viscometer following to [36].

### 2.2.3. Fiber Spinning

The fibers were spun by the mechanotropic method [37] on a laboratory spinning line, where the relative humidity and the temperature were set at 20 wt% and 25 °C, respectively. The method involves stretching a highly viscous solution jet in the air with high elongation ratios, leading to the onset of phase separation, and the solvent is released from the center of the jet to the surface of the spinning fiber. In the case of the humid air, it is possible that the superposition of two mechanisms causes polymer separation from solution jets: under diffusion of the air moisture in, and due to strong extension leading to the phase separation and selection of the solvent out [37–39]. To feed the solution, a Malvern Rosand RH10 capillary rheometer was used, while a monofilament spinneret with a hole diameter of 500 μm was used to control the solution flow rate, set at 0.08 m/min. The spun fiber then has two stages of orientation stretching: in air, followed by washing with water to remove any residual solvent on the fiber surface, drying at 80 °C, and finally plasticized as-spun fibers—the thermal drawing at 120 °C. The maximum possible draw ratio for each copolymer was achieved at each stage by controlling a speed that allowed for stable spinning without breakage, resulting in the most oriented fiber. Figure 1 displays the scheme of the spinning line.

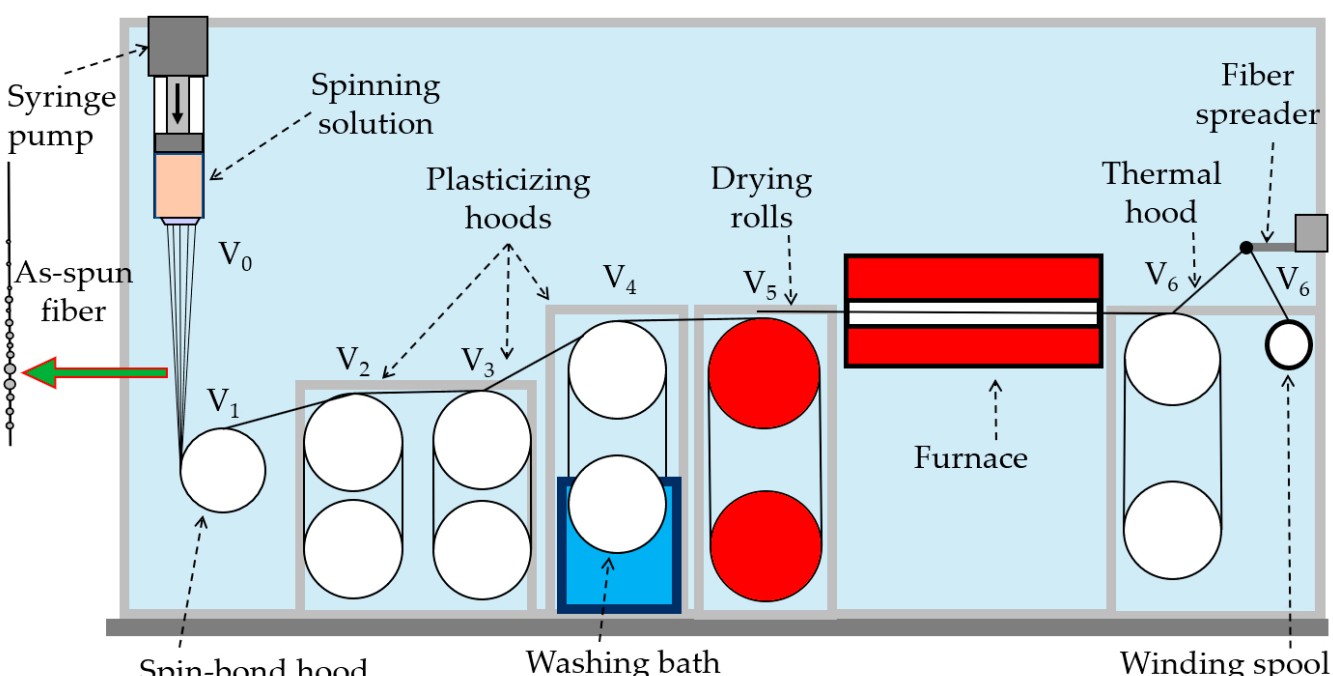

**Figure 1.** Scheme of the mechanotropic spinning line. $V_0$ is the linear flowing speed from the die, and $V_1$–$V_6$ are the winding speeds.

### 2.2.4. Fiber Characterization

The diameter of each fiber was averaged from at least ten measurements along different locations on the fiber. To perform these measurements, the Biomed 6PO optical microscope coupled with a Touptek XFCAM1080PHD camera with a magnification of 60×, which has an accuracy of ±0.3 μm, was used. The inhomogeneity was characterized by the difference between the maximum and minimum values of the fiber diameter.

The mechanical properties of the fibers were measured using an Instron 1122 tensile machine, with a basic filament length of 10 mm. All measurements were performed at 23 ± 2 °C. The extension speed was 10 mm/min. The reported results are averaged for at least 10 tests.

## 3. Results and Discussion

### 3.1. Solutions' Rheology

First of all, the behavior of dilute solutions of different copolymers in DMSO was examined. The intrinsic viscosity values were determined, and the Huggins constants [40] were calculated using the equation $\eta_{sp}/c = [\eta] + K_H[\eta]^2c$. Additionally, the Martin constants [41] were calculated using the equation $\ln(\eta_{sp}/c) = \ln([\eta]) + K_M[\eta]c$. Figure 2 illustrates the relationship in Huggins coordinates.

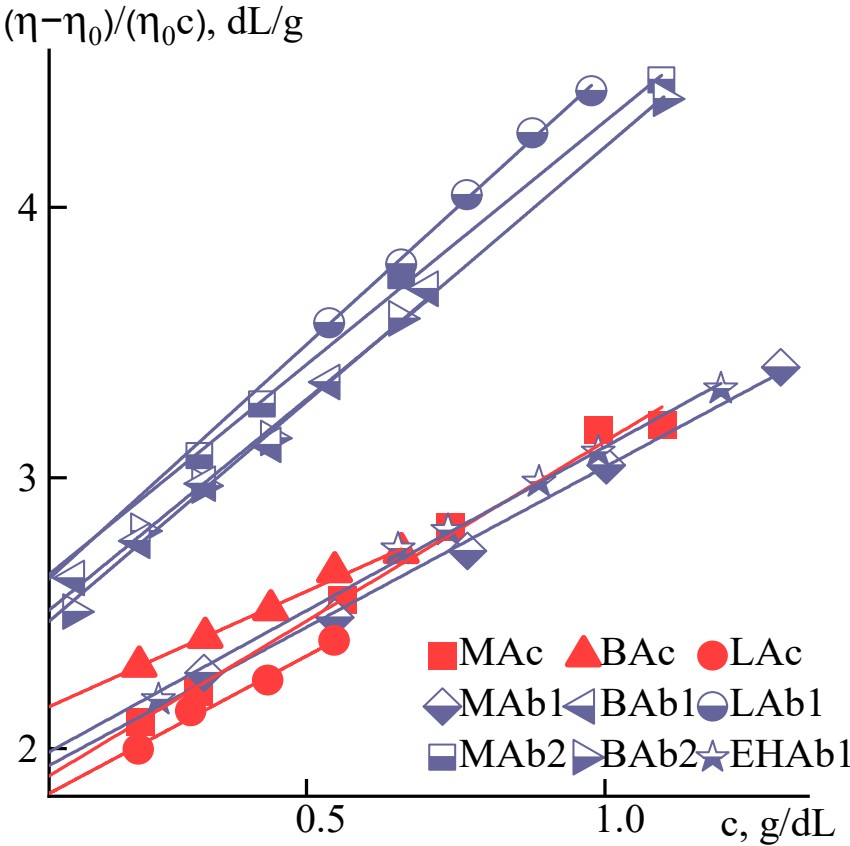

**Figure 2.** Concentration dependence of viscosity in coordinates of Huggins equation.

In the case of flexible-chain polymers, the value of $K_H$ increases as the solvent quality gets worse, indicating that polymer–polymer interactions become more favorable than polymer–solvent interactions [42,43]. It was observed that polymers with high intrinsic viscosities could be obtained for all samples with the simultaneous addition of components under the same synthesis conditions, irrespective of the type of alkyl acrylate used. Such polymers exhibit slightly better solvent affinity, which is reflected in the lower values of Huggins and Martin constants compared with continuous reagent loading. The properties of dilute solutions of these polymers are summarized in Table 2.

**Table 2.** Intrinsic viscosity values, $K_H$, and $K_M$ parameters of the copolymers.

| Sample | [η], dL/g | $K_H$ | $K_M$ |
|--------|-----------|-------|-------|
| MAc | 1.8 | 0.4 | 0.3 |
| MAb1 | 1.9 | 0.3 | 0.2 |
| MAb2 | 2.5 | 0.3 | 0.2 |
| BAc | 2 | 0.2 | 0.2 |
| BAb1 | 2.4 | 0.3 | 0.2 |
| BAb2 | 2.3 | 0.3 | 0.2 |
| LAc | 1.8 | 0.4 | 0.3 |
| LAb1 | 2.5 | 0.3 | 0.2 |
| EHAb1 | 1.9 | 0.3 | 0.2 |

By considering these data, a generalized relationship between reduced coordinates of Martin equation viscosity and the volume occupied by macromolecule concerning polymer–solvent interaction was derived (Figure 3). This dependence confirms that the flow behavior of dilute solutions is mainly influenced by the intensity of the interaction between the polymer and solvent and the volume of the macromolecular coils present in the solution. These results are consistent with those reported previously for copolymers containing acryl amide [44].

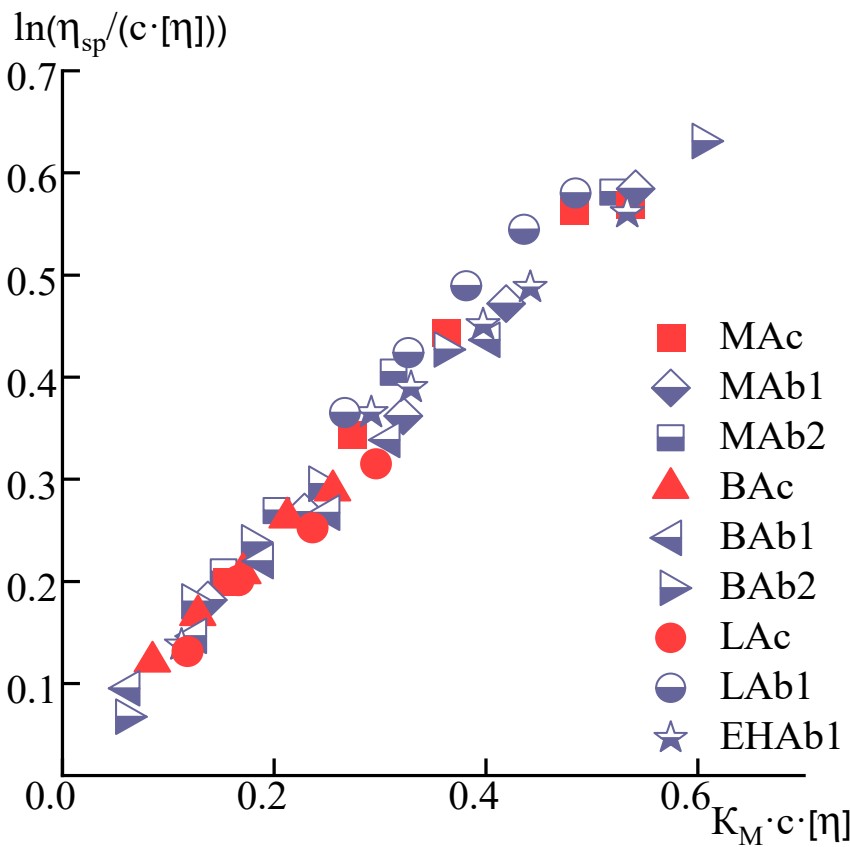

**Figure 3.** The relationship between the reduced viscosity and the volume of macromolecules present in the solution using the Martin equation coordinates.

As the concentration of the solutions increases in the region of concentrated solutions, which is more desirable for stable spinning and minimizing solvent usage, the behavior of solutions containing different copolymers begins to change. Figure 4 displays the flow curves of a series of 30 wt% solutions at 70 °C.

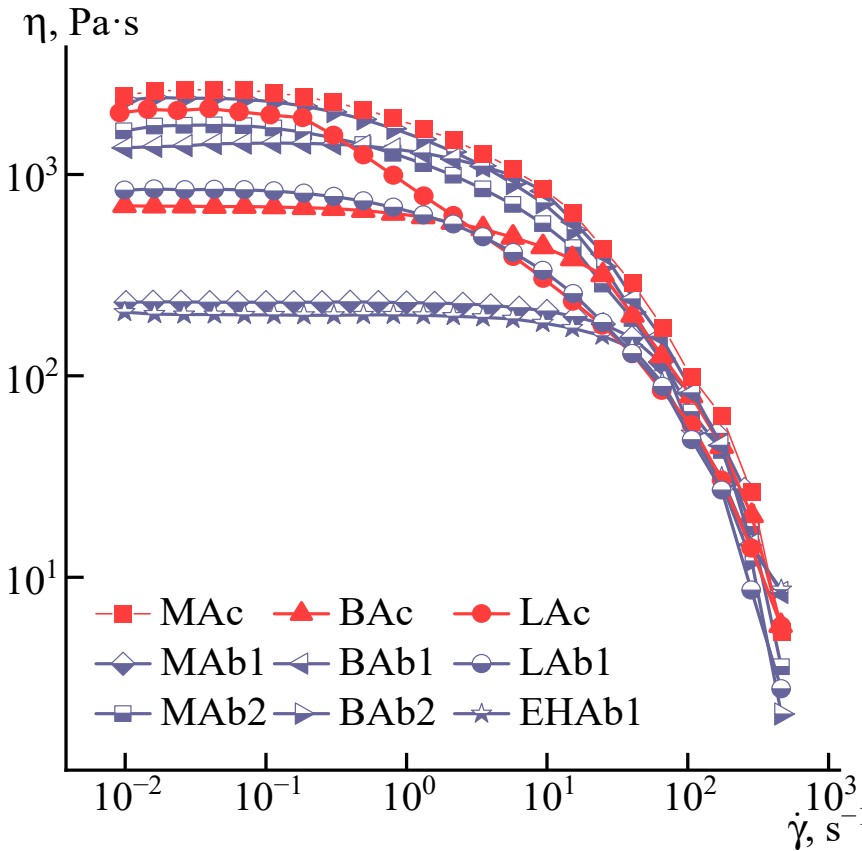

**Figure 4.** Flow curve of 30 wt% PAN terpolymers' solutions at 70 °C.

The non-Newtonian flow behavior of solutions of lauryl acrylate copolymers is particularly evident in copolymers with higher molecular weights and predominantly has nonuniform distribution of comonomers along the chain (LAc). At high concentrations, the values of the highest Newtonian viscosity of solutions vary significantly for different solutions at the same concentration of polymers, and the expected correlation between viscosity and intrinsic viscosity of solutions is not observed. These differences indicate that, in addition to the molecular weight of the polymer, intermolecular polymer–polymer interactions, which increase in concentrated solutions, are the decisive factor.

Viscoelastic properties of the solutions were analyzed using the rotational rheometry method under linear viscoelasticity conditions (at 1% of deformation). Figure 5 presents the results obtained for 30% solutions at 70 °C.

The structuring of the samples is primarily determined by the density of the entanglement network in the solution, which depends on the polymer concentration and the volume of macromolecules in a solution, proportional to the intrinsic viscosity. A strong correlation between viscoelastic properties and steady-state shear rheology data was observed in the studied series. Samples with a predominantly nonuniform distribution of comonomers exhibit higher values of the viscosity and loss modulus with a lower slope angle of the frequency dependences in the terminal zone.

The copolymers with lauryl acrylate exhibit the highest degree of structuring in the low-frequency region. In a series of nonuniform copolymers, the behavior of a sample with an ethylhexyl group obeys the Maxwell model [45]. Among copolymers with uniform distribution, the copolymer with methyl acrylate, which has the lowest molecular weight, has the least deviation from the Maxwell dependence in the low-frequency range.

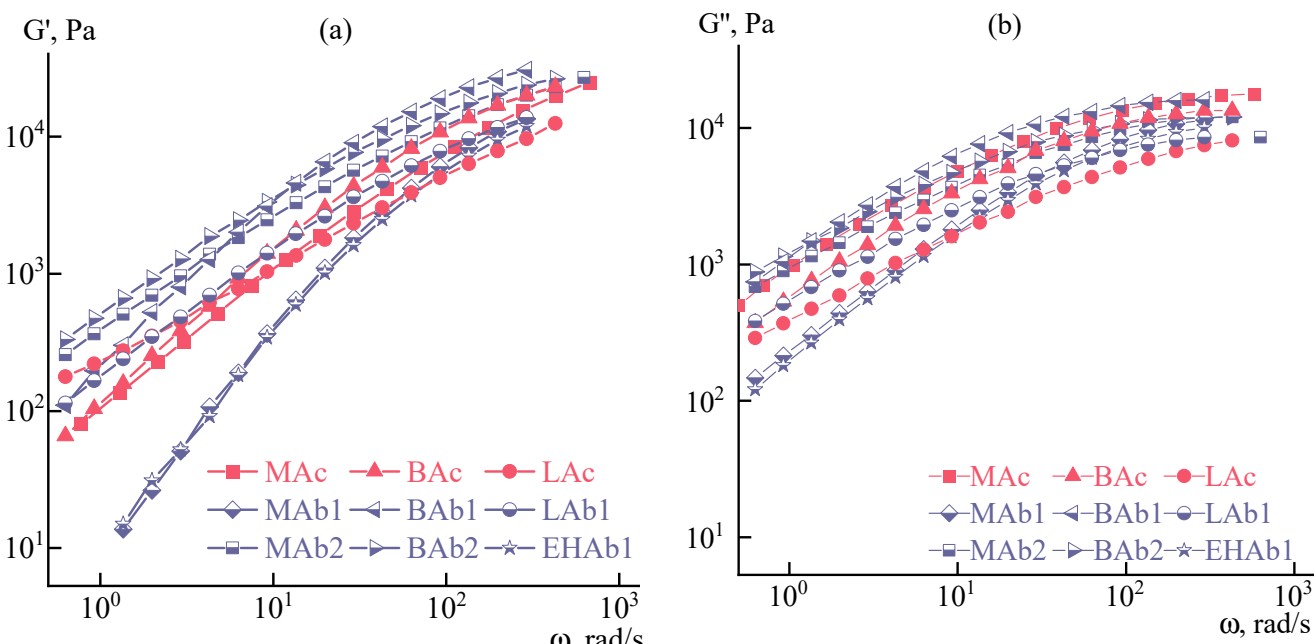

**Figure 5.** Dependences of elastic (**a**) and loss moduli (**b**) on the frequency of 30 wt% PAN solutions at 70 °C.

### *3.2. Fiber Spinning*

The effect of spinneret drawing (drawing between the spinneret and the first winding roller), as well as subsequent drawing stages on the fiber mechanical properties, has been investigated using as an example of spinning of MAb1 solution.

### 3.2.1. Spinneret Drawing Influence

The data on the influence of spinneret drawing on the mechanical properties of the fiber and its diameter are presented in Figure 6.

Increasing the draw ratio at this stage primarily affects the fiber diameter. This is because the solution jet exiting the die is a viscoelastic liquid and undergoes viscous deformation until the moment of phase separation onset, which is caused by the combined action of high uniaxial deformation and air moisture. The mechanical properties of the fiber, particularly the elongation at break, change non-linearly as the spinneret drawing rate increases. In the relatively low drawing ratio regime (up to 40 for the solution jet under study), large-diameter fibers are obtained with extremely low strength and very low elongation at break. These properties are possible to explain by prevailing of filament coagulation by air moisture at weak extension. The resulting opaque fiber (Figure 7) contains interfacial borders and, presumably, defects caused by fast diffusions of the water vapor, similar to those observed during wet fiber spinning [46,47], or membrane obtained in a humid environment [48–50].

An increase in the drawing ratio of the solution jet beyond 40 results causes a shift in the mechanism of phase separation from coagulation to strain-induced. In this case, the phase separation occurs throughout the volume of the jet with the achieved separation of polymer phase being in the center, while that achieved by the solvent phase being squeezed in the outward direction, ultimately resulting in the formation of a homogeneous and dense fiber [51,52]. As a result, the elongation at the break of the fiber increases to ~100%, the strength increases by 4 times, and the fiber diameter decrease to ~20 μm, reaching an almost constant value that weakly depends on a subsequent increase in the drawing ratio. This behavior is explained by the fact that the solid fiber formed after phase separation behaves as a single unit during uniaxial tension, becoming resistant to further stretching.

Based on the experiment, the spin draw ratio was selected for all samples to achieve a fiber diameter of approximately 30 μm, which is sufficient for subsequent orientational drawing to obtain a final fiber of 10–15 μm. Moreover, the phase separation in the stretchable solution jet mainly occurs due to the high deformation mechanism.

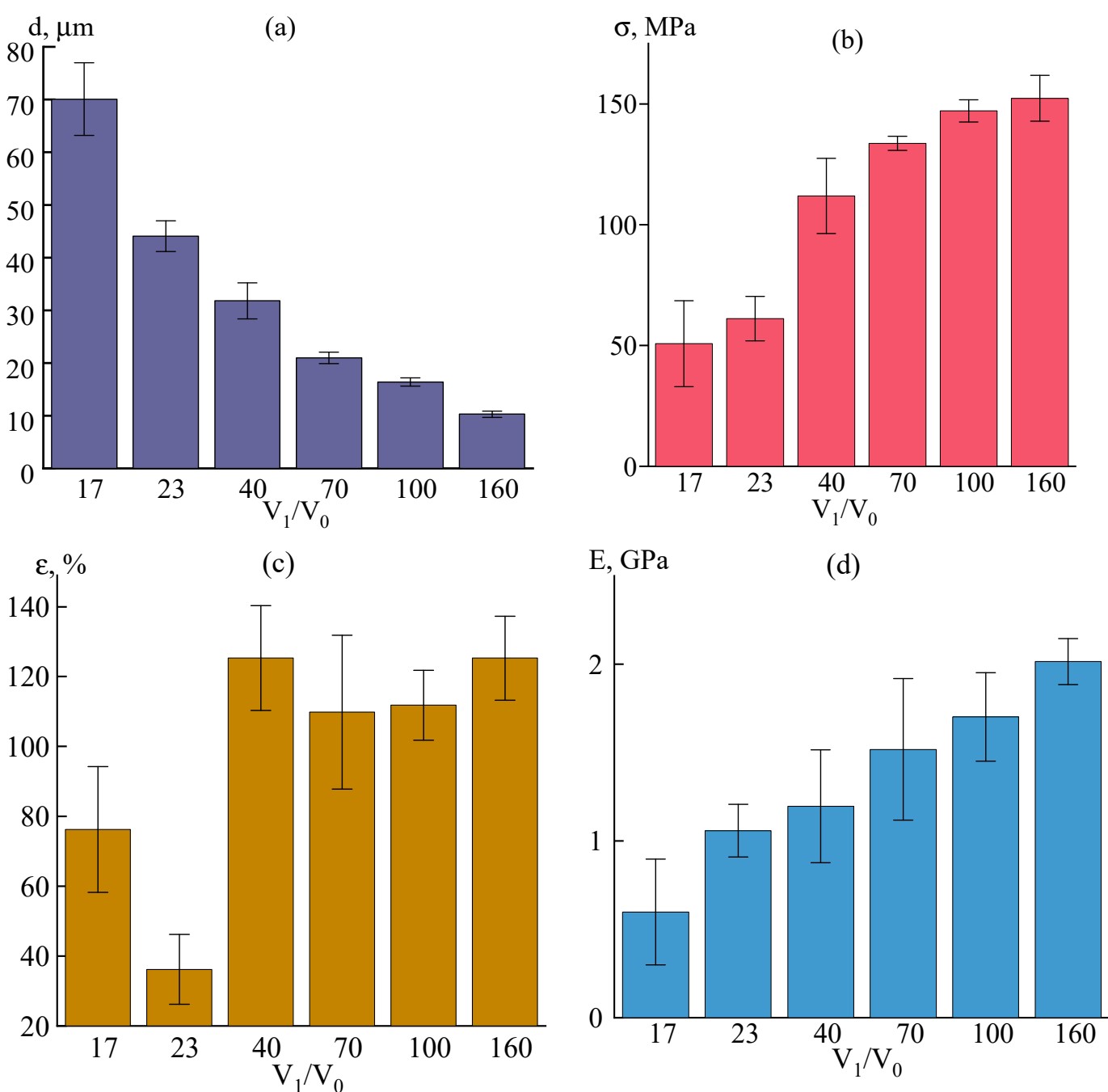

**Figure 6.** Dependences of (**a**) diameter, (**b**) strength, (**c**) elongation at break, and (**d**) modulus of elasticity for fibers obtained by stretching ratio of the MAb1 solution in the first stage of the fiber spinning (see Figure 1).

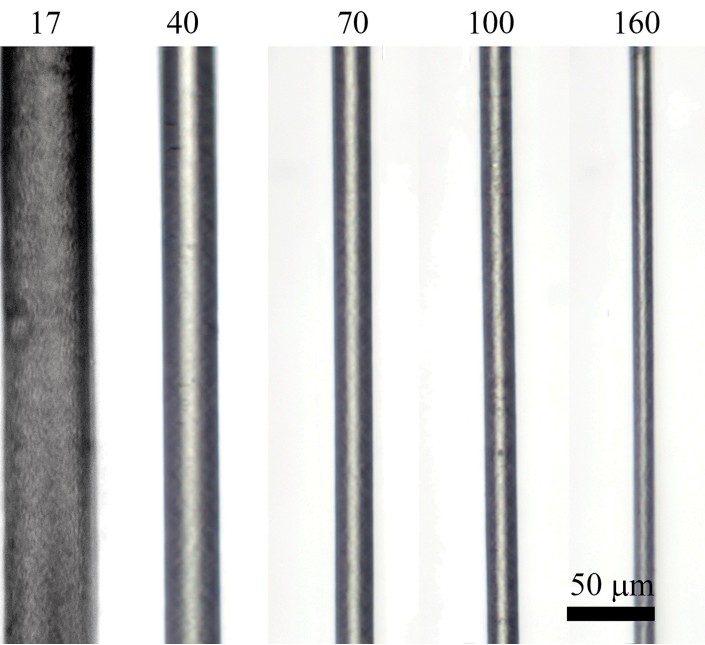

**Figure 7.** The morphology of the MAb1 fiber from the first drawing stage at various $V_0/V_1$ ratios marked above the fibers images.

### 3.2.2. Influence of Orientation Drawing

The structure and properties of the PAN fibers obtained by wet and dry–wet spinning are greatly affected by the subsequent orientation and plasticization stages of stretching [53,54]. The subsequent stages of fiber spinning by mechanotropic method were performed under conditions of maximum stretching at each stage, preserving a continuous process without breaking the spinning fiber. The evolution of fiber diameter and its mechanical properties at various stretching stages were investigated using as an example a fiber from MAb1 copolymer prepared at a spinneret draw ratio of 40.

Figure 8 shows images of the fibers obtained.

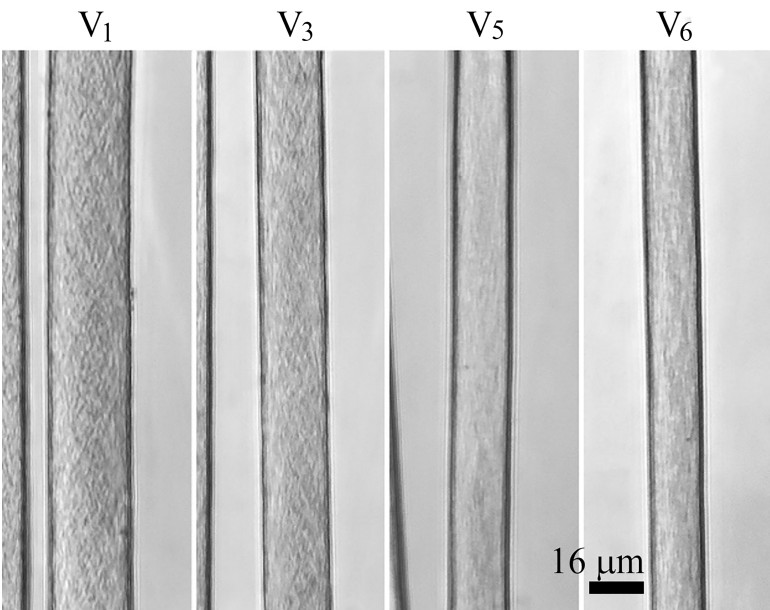

**Figure 8.** Images of the obtained fibers from different stages of spinning at $V_1 = 4.2$, $V_2 = 5.7$, $V_5 = 12.5$, $V_6 = 17.5$ m/min.

The diameter of the fiber decreases by ~2 times after passing through all stages of drawing, resulting in a transparent fiber without pronounced fibrillation or defects. Figure 9 shows the mechanical properties at different stages of the spinning process.

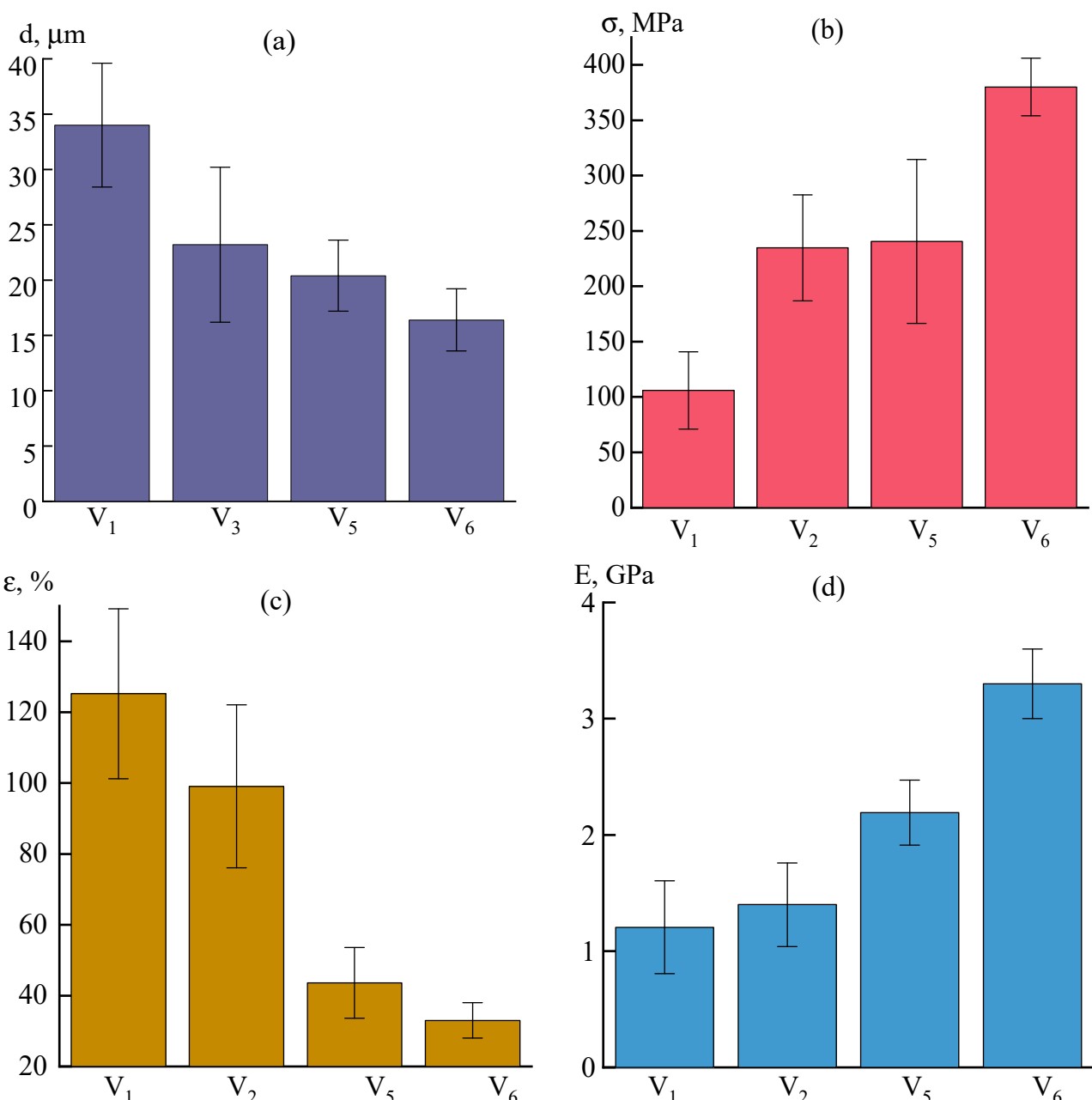

**Figure 9.** Values of diameter (**a**), strength (**b**), elongation at break (**c**), and modulus of elasticity (**d**) for MAb1 fibers after subsequent stages of drawing: $V_1$, $V_2$, $V_5$, and $V_6$.

The strength and modulus of elasticity increase by approximately 3.5 times as the fiber is drawn. The main stretching and strengthening processes occur at the initial stages of orientation hood ($V_1$–$V_5$), followed by additional stretching at the thermal hood stage ($V_6$) with a corresponding decrease in elongation at break from 123% to 40%, and then from 40% to 30%, respectively. The developed process conditions allowed for continuous spinning without any breaks during the experiment (more than 5 h of continuous spinning from a one-hole spinneret). An example of a spool with spun fibers is shown in Figure 10.

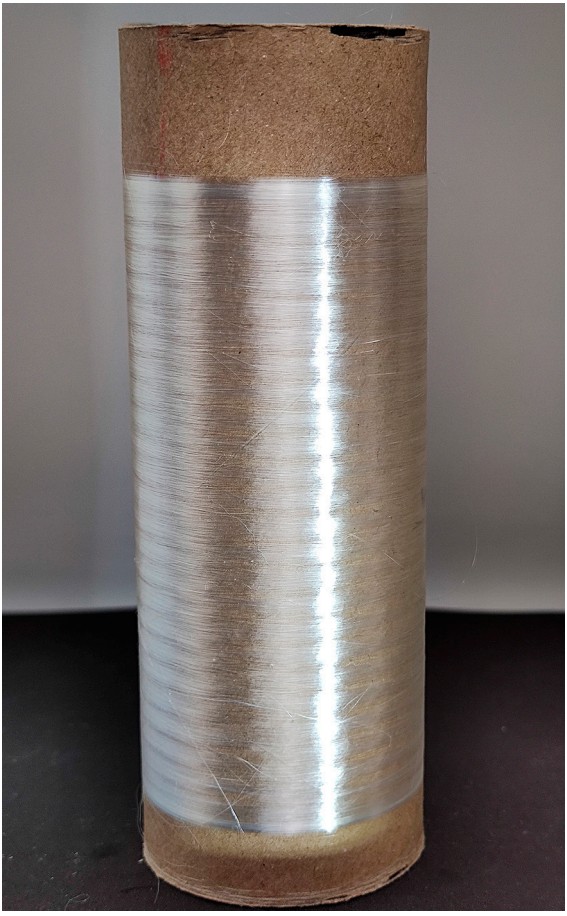

**Figure 10.** Spool with MAb1 spun fiber.

Furthermore, fibers were obtained from all other copolymer solutions by using the maximum possible drawing ratios at each stage. The detailed conditions for their formation are provided in Table 3.

**Table 3.** Solutions' spinning parameter.

| Title 1 | Linear Flow ($V_0$) and Winding ($V_1$–$V_6$) Speed, m/min | | | | | | Draw Ratio | |
| --- | --- | --- | --- | --- | --- | --- | --- | --- |
| | $V_0$ | $V_1$ | $V_2$ | $V_3$ | $V_4, V_5$ | $V_6$ | $V_6/V_1$ | $V_6/V_0$ |
| MAc | | 3.8 | 6.1 | 10 | 10 | 20 | 5.3 | 250.0 |
| MAb1 | | 4.2 | 5.7 | 10.4 | 12.5 | 17.5 | 4.2 | 218.8 |
| BAc | 0.08 | 4.4 | 10.4 | 12.2 | 12.3 | 21.9 | 5.0 | 273.8 |
| BAb1 | | 2.2 | 10.1 | 11.9 | 13 | 21.2 | 9.6 | 264.4 |
| LAc | | 2.2 | 6.8 | 7.1 | 7.1 | 16.3 | 7.4 | 203.8 |
| LAb1 | | 2.2 | 10 | 12.5 | 12.9 | 17.3 | 7.9 | 216.3 |

The variation of process parameters is due to differences in the behavior of the jet during stretching under conditions of spinneret drawing and subsequent orientation and thermal drawings. It should be noted that for all systems, the highest draw ratio is achieved in the first stages of orientation drawing ($V_2$ and $V_3$), and the resulting fiber stretches well when heated above the glass transition temperature of PAN during the thermal drawing stage, allowing for application of double stretching ratio of the fiber. The comparative mechanical properties of the resulting fibers, prepared at various modes of component loading at synthesis, are shown in Figure 11.

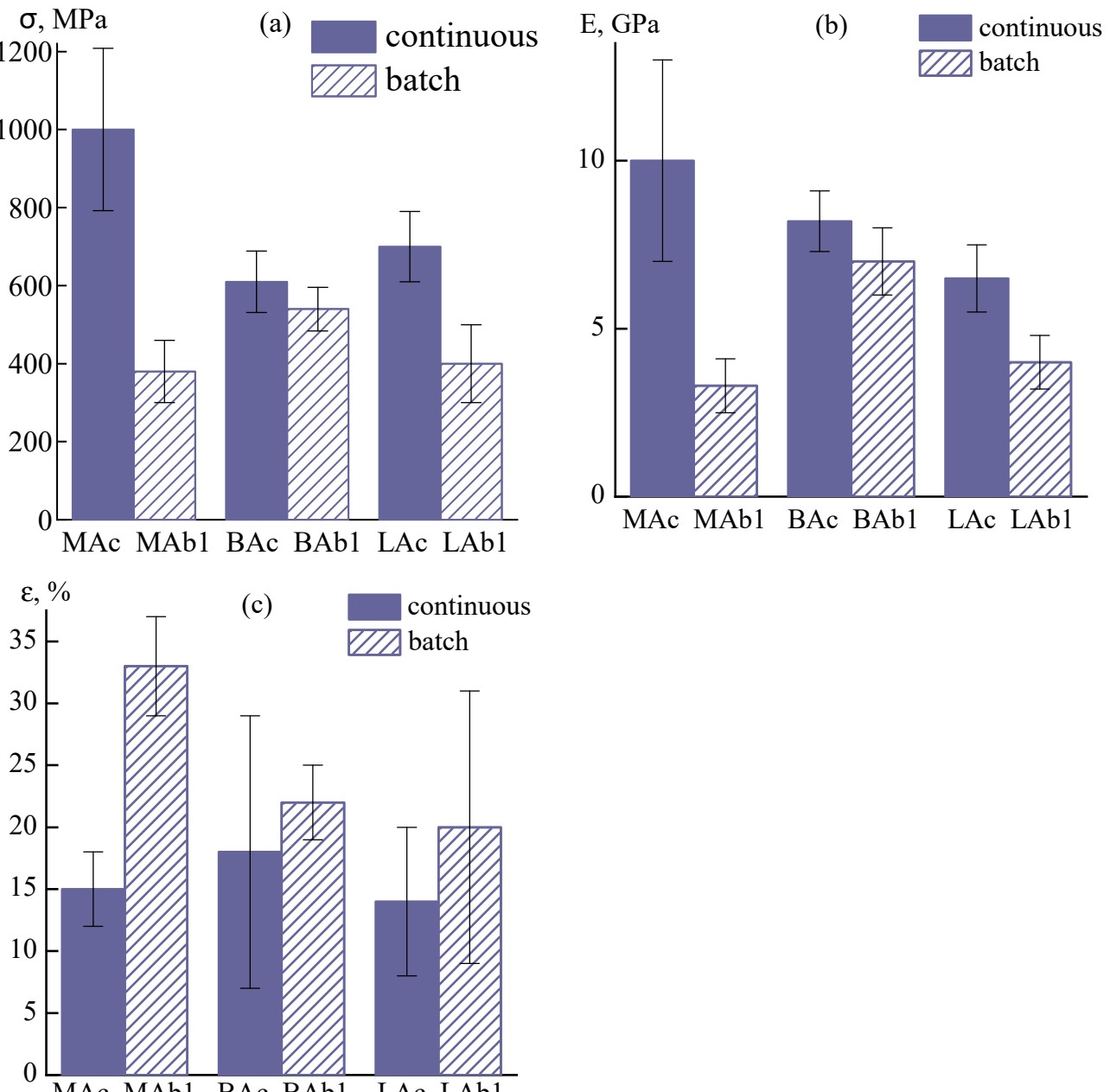

**Figure 11.** Strength (**a**), modulus of elasticity (**b**), and elongation at break (**c**) for fibers obtained from polymers synthesized by the continuous and one-time introduction of monomers modes.

Solutions with a more uniform comonomers' distribution despite the most severe orientation and plasticizing stretching, resulting in lower mechanical properties. Maybe, the additional thermal stretching steps are necessary to produce high-strength fibers. However, as per used processing stages, the best mechanical properties for all copolymers were observed in nonuniform copolymers synthesized by continuously introducing comonomers. Among the series, copolymers containing methyl acrylate achieved the highest residual elongation at break values. Fibers spun from copolymers with methyl acrylate have the maximum strength (over 1000 MPa) and elastic modulus of 10 GPa at a relative elongation at a break of 15%.

## 4. Conclusions

The order of monomers introduction to the reaction mixture during copolymer synthesis directly affects their rheological properties in the concentrated solution region where intermolecular interactions become significant. In this case, the nature of unit distribution contributes significantly to the properties of solutions and spun fibers more than the typical nature of alkyl acrylate used. Deviations from this behavior were observed only in the LA series, where the presence of lauryl acrylate caused a change in the flow of concentrated solutions, presumably, due to its poor solubility stipulated by the long aliphatic substituents. Solution jets of copolymers with a uniform distribution of alkyl acrylate units along the chain exhibit the poorest stretching ability in the air. The same concerns fibers stretching at temperatures below the PAN glass transition, as shown for the entire series studied. The best mechanical properties are achieved by adding methyl acrylate, which produces the most durable and high-modulus fibers in the case of a nonuniform copolymer.

**Author Contributions:** Conceptualization, I.Y.S. and E.V.C.; methodology, M.S.K., I.Y.S. and L.A.V.; validation, I.Y.S., V.G.K., E.V.C. and G.K.S.; investigation, M.S.K., A.F.V., R.V.T. and L.A.V.; data curation, I.Y.S.; writing—original draft preparation, I.Y.S., M.S.K. and L.A.V.; writing—review and editing, I.Y.S. and V.G.K.; supervision, V.G.K., E.V.C. and G.K.S.; project administration, V.G.K.; funding acquisition, V.G.K. All authors have read and agreed to the published version of the manuscript.

**Funding:** Rheological properties of compositions were funded by Russian Science Foundation grant number 17-79-30108. A fiber spinning investigation was carried out within the State Program of TIPS RAS.

**Data Availability Statement:** The data that support the findings of this study are available from the corresponding author upon reasonable request.

**Acknowledgments:** This work was performed using the equipment of the Shared Research Center "Analytical center of deep oil processing and petrochemistry of TIPS RAS".

**Conflicts of Interest:** The authors declare no conflict of interest.

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
