# Peer review of "Fiber Spinning of Polyacrylonitrile Terpolymers Containing Acrylic Acid and Alkyl Acrylates"

_fibers, doi:10.3390/fib11070065_

Round 1

Reviewer 1 Report

The authors presented a detailed study on various factors (e.g., alkyl acrylates with different chain lengths added in the copolymer solutions, drawing speed and temperature) impacting the mechanical properties of the polymeric fibers produced via the mechanotropic method. Especially valuable is the study on the diameters and the mechanical properties of the fibers in different drawing stages, which is important for understanding/controlling the quality of the final product. I recommend the publication of the manuscript after the authors considered the following comments and suggestions.

1. All the data presented in Figures 2-6, 9 and 11 do not have error bars showing their uncertainty of the measurements. The authors need to include a standard deviation for each data point.

2. The data for elongation at break in Figure 6c requires more discussion. First of all, showing an error bar for each data column is important here so that the reader will know whether there is a possibility that the elongation at break plateaus beyond V1/V0=40. Another thing for the authors to consider is to examine whether the behaviour observed in Figure 6c is dependent on the extension rate (when measuring elongation at break).

3. The authors should check the Figure 2 caption, which appears to be strange.

Author Response

Dear colleague,

Thank you for taking the time to review our publication and for your thorough analysis. We have carefully considered all of your questions and provided detailed responses below.

The authors presented a detailed study on various factors (e.g., alkyl acrylates with different chain lengths added in the copolymer solutions, drawing speed and temperature) impacting the mechanical properties of the polymeric fibers produced via the mechanotropic method. Especially valuable is the study on the diameters and the mechanical properties of the fibers in different drawing stages, which is important for understanding/controlling the quality of the final product. I recommend the publication of the manuscript after the authors considered the following comments and suggestions.

  1. All the data presented in Figures 2-6, 9 and 11 do not have error bars showing their uncertainty of the measurements. The authors need to include a standard deviation for each data point.

The results really should be presented with error bars.

However, capillary viscometry data have very good reproducibility (several seconds with a total flow time of 200-600 seconds. Thus, error bars do not exceed the size of a point on the graph. The same applies to rotational rheometry data, where data are presented in logarithmic scales in the range of 3-4 orders of magnitude. Therefore, usually, rheological data are given without error bars, so as not to download graphs.

Error bars have been added in Fig. 6, 9, and 11.

  1. The data for elongation at break in Figure 6c requires more discussion. First of all, showing an error bar for each data column is important here so that the reader will know whether there is a possibility that the elongation at break plateaus beyond V1/V0=40. Another thing for the authors to consider is to examine whether the behaviour observed in Figure 6c is dependent on the extension rate (when measuring elongation at break).

Error bars have been added.

Indeed, the elongation tends to reach a plateau, which is associated with the achievement of some equilibrium deformation of the jet before phase separation, as written in the text. For the best readability, the mechanical properties and fiber morphology images have been placed by each other followed the full description of the data obtained.

We did not study the effect of the tensile rate at the break on the mechanical properties of the samples, while there should be a certain effect, so we carried out measurements under the same conditions specified in the methodological part. Here, the observed transition between 23 and 40 V1/V0 ratios seems important, apparently due to the transition from the coagulation mechanism of jet hardening to the deformation one. The later leads to the formation of a more monolithic structure and a less number of defects caused by deposition of air moisture as in the coagulation case. But in this work, we can judge this situation indirectly, because did not carry out the high-resolution morphology and structure of such fibers since this phenomenon has not been the main part of the publication.

  1. The authors should check the Figure 2 caption, which appears to be strange.

Thank you, indeed the legend remained from the template, in the current version it has been changed.

Reviewer 2 Report

Attached

Good

Author Response

Dear colleague,

Thank you for taking the time to review our publication and for your thorough analysis. We have carefully considered all of your questions and provided detailed responses below.

The present manuscript examines the Fiber Spinning of Polyacrylonitrile Terpolymers Containing Acrylic Acid and Alkyl Acrylates. Overall, the quality of this manuscript is good but with edits and changes, I believe this work should almost be ready for publication in “Fibers” journal.

  1. Emphasize the novelty of this work in the abstract section.

The novelty of the work was emphasized in the abstract (following the Abstract limit):

Terpolymers of acrylonitrile with acrylic acid and alkyl acrylates, including methyl-, butyl-, 2-ethylhexyl-, and lauryl acrylates, were synthesized using the reversible addition-fragmentation chain transfer method. In this study, the focus was on the investigation of the impact of different monomer addition methods (continuous and batch) on both the rheological behavior of the spinning solutions and the mechanical properties of the resulting fibers. Our findings revealed that the method of monomer addition, leading either to non-uniform copolymers or to a uniform distribution, significantly influences the rheological properties of the concentrated solutions, surpassing the influence of the alkyl-acrylate nature alone. To determine the optimal spinning regime, we examined the morphology and mechanical properties at different stages of fiber spinning, considering spin-bond and orientation drawings. The fiber properties were found to be influenced by both the nature and introducing method of the alkyl-acrylate comonomer. Remarkably, the copolymer with methyl acrylate demonstrates the maximum drawing ratios and fiber tensile strength, reaching  1 GPa. Moreover, we discovered that continuous monomer addition allows for reaching the higher drawing ratios and superior fiber strength compared to the batch method.

  1. Consider dividing the second keyword into two separate terms: "reversible additionfragmentation" and "chain-transfer polymerization."

RAFT (Reversible Addition Fragmentation chain-Transfer) polymerization is a single term.

  1. Some keywords are excessively long, such as "mechanical properties of the fibers." It would be more concise to use "mechanical properties" instead. Limit the number of keywords to 5 or 6.

The keywords were rewritten: co-polyacrylonitrile; RAFT process; comonomer sequence; alkyl acrylate; mechanotropic spinning; fibers; mechanical properties

In the Fibers Journal is possible to use up to 10 keywords: (List three to ten pertinent keywords specific to the article yet reasonably common within the subject discipline.)

  1. Include the names of manufacturers and countries for each material or chemical in subsection 2.1.

Detail synthesis parameters and polymer characterization are given in detail in previous paper [28]. The chemical manufacturers have been added.

  1. Ensure proper labeling of the Y and X-axes for every figure. For instance, follow the example below:

All labels have been checked according to MDPI rules.

I suggest a major revision on the above-mentioned issues before publication, which could

improve the quality of the manuscript further. Goodluck.

Reviewer 3 Report

The manuscript presents an experimental study on wet-spinning and properties of various PAN co-polymers including terpolymers. Authors begin by stating that “Although the process of PAN fiber carbonization has been studied for a long time [3, 4], the optimal copolymer composition required for producing high-quality precursor fibers and high-strength carbon fibers from them remains uncertain.” However, the manuscript presents NO results on resulting carbon fibers. Therefore, the initial goal of the work remains unclear.

As for composition, various monomers and terpolymers of PAN have been reported in the literature (some cited by the authors for wet-spinning), but not cited for melt-spinning (viz. T. Mukundan, et al., “A photocrosslinkable melt processible acrylonitrile terpolymer as carbon fiber precursor”, Polymer, 47, 4163-4171, 2006

N

Author Response

Dear colleague,

Thank you for taking the time to review our publication and for your thorough analysis. We have carefully considered all of your questions and provided detailed responses below.

The manuscript presents an experimental study on wet-spinning and properties of various PAN co-polymers including terpolymers. Authors begin by stating that “Although the process of PAN fiber carbonization has been studied for a long time [3, 4], the optimal copolymer composition required for producing high-quality precursor fibers and high-strength carbon fibers from them remains uncertain.” However, the manuscript presents NO results on resulting carbon fibers. Therefore, the initial goal of the work remains unclear.

We have corrected the introduction because the main goal of the work was to obtain precursors, but not carbon fibers. The abstract has been rewritten and supplemented:

The goal of this study was the investigation of the influence of the addition of a broad range of ternary copolymers of PAN with various alkyl acrylates, including methyl-, butyl-, ethylhexyl-, and lauryl acrylate on the solution rheology, novel mechanotropic fiber spinning process without using of coagulation baths and final fiber properties.

As for composition, various monomers and terpolymers of PAN have been reported in the literature (some cited by the authors for wet-spinning), but not cited for melt-spinning (viz. T. Mukundan, et al., “A photocrosslinkable melt processible acrylonitrile terpolymer as carbon fiber precursor”, Polymer, 47, 4163-4171, 2006

Indeed, the development of copolymers and the search for suitable plasticizers suitable for the production of PAN fibers from melts have been underway since the 60s.

Melt spinning is quite different from solution spinning, so we did not include the articles on melt spinning of PAN in the Introduction, though earlier some coauthors of this paper published the review entitled “Melt-Spinnable Polyacrylonitrile—An Alternative Carbon Fiber Precursor” (doi.org/10.3390/polym14235222).

Round 2

Reviewer 2 Report

NA

Reviewer 3 Report

Authors have appropriately deleted their goal of using their fibers as carbon fibre precursors, so the results are now just those for PAN-based fibers.